# Iron-Loaded Catalytic Silicate Adsorbents: Synthesis, Characterization, Electroregeneration and Application for Continuous Removal of 1-Butylpyridinium Chloride

**Imen Ouiriemmi, Aida M. Díez \***[ID]**, Marta Pazos**[ID] **and María Ángeles Sanromán**[ID]

CINTECX, Universidade de Vigo, Chemical Engineering Department Campus As Lagoas-Marcosende, University of Vigo, 36310 Vigo, Spain; imenwrimi@hotmail.com (I.O.); mcurras@uvigo.es (M.P.); sanroman@uvigo.es (M.Á.S.)

**\*** Correspondence: adiez@uvigo.es; Tel.: +34-986-812304

**Abstract:** This research proposes the application of iron-loaded sepiolite (S-Fe) as a catalytic adsorbent for the unreported 1-butylpyridinium chloride ([bpy] Cl) treatment in an aqueous medium. Initially, sepiolite was selected as an inexpensive and efficacious adsorbent for [bpy] Cl elimination. After that, sepiolite was loaded with iron for the subsequent electro-Fenton (EF) regeneration treatment. Once kinetic and isotherm studies were performed, providing respectively almost instantaneous adsorption (20 min) and an uptake of 22.85 mg/g, [bpy] Cl adsorption onto S-Fe was studied in continuous mode. The obtained breakthrough curve was analyzed using three standard breakthrough models, being Yoon–Nelson and Thomas the most suitable adjustments. Afterwards, S-Fe regeneration by the EF process was conducted using this iron-loaded silicate material as a heterogeneous catalyst. Under optimized operational conditions (current intensity 300 mA and $Na_2SO_4$ 0.3 M), complete adsorbent regeneration was achieved in 10 h. The total mineralization of [bpy] Cl was reached within 24 h and among seven carboxylic acids detected, oxalic and acetic acids seem to be the primary carboxylic acids produced by [bpy] Cl degradation. Finally, S-Fe was efficiently used in four consecutive adsorption–regeneration cycles without a noticeable reduction in its adsorption capacity, opening a path for future uses.

**Keywords:** adsorption; electro-Fenton; ionic liquid; regeneration; fixed-bed column

## 1. Introduction

Ionic liquids (ILs) are complex salts [1], which, due to their high thermal and chemical stabilities, can accrue in the environment [2]. Moreover, some biological studies have demonstrated the toxicity of these pollutants in the environment [3]. The potential usages of these compounds make them be widely researched and potentially used at industrial scale. Thus, their apparition on wastewater is a matter of time [2]. In the literature, several studies proved the toxicity of ILs to aquatic organisms and highlighted the real cause for concern [4,5]. Besides, the efficiency of the existing biological technologies to remove ILs is limited due to their poor biodegradability and high stability [6]. In this line, the overall objective of this study is to systematically gather information on the behavior of ILs by conventional treatments and design an eco-efficient strategy for their removal.

Among the numerous conventional techniques that have been proposed for the handling of wastewater contaminated with organic pollutants, adsorption has been highlighted. This process's advantages include its rapidity, easy set-up and great effectiveness. Nevertheless, the regeneration of the adsorbent and its cost are the essential defects of this technique [7].

To minimize the operational costs of the adsorption process, several low-cost adsorbents have been employed for wastewater decontamination. Among these adsorbents, natural adsorbents such as Illite and Montmorillonite [7], silicate-based materials, which are widely spread and facile to collect [8,9], or sludge-based adsorbents have been used instead of the efficient but expensive activated carbon [10].

The exhausted adsorbents' regeneration and their subsequent reuse may decrease the adsorption treatment cost. Moreover, regeneration may decrease the problems associated with exhausted adsorbents disposal [7]. Usually, solvent extraction and/or thermal regeneration are used. Nevertheless, the former implies the addition of other agents (pH or salt content modifications), and in the case of strong interactions between the adsorbate and adsorbent, the removal percentages are below acceptable levels [10]. Alternatively, the thermal treatment may generate some toxic gases, damage the structure of the adsorbent and, even in this case, complete regeneration might not be achieved [11].

As reported in previous research [7,12], the regeneration of the adsorbent can be successfully performed via advanced oxidation processes (AOPs). These are based on the production of hydroxyls radicals ($\cdot$OH) (2.7 V vs SHE) [13]. $\cdot$OH may be produced by several ways such as direct anodic oxidation (AO), the Fenton process (by the reaction of $H_2O_2$ and $Fe^{2+}$) or the electro-Fenton (EF) process which generates and regenerates, respectively, $H_2O_2$ and $Fe^{2+}$, thus avoiding the constant reagent addition necessary in the Fenton process [13].

Applying the EF process when using immobilized iron into the adsorbent, avoids the generation of iron sludge when compared to the usage of homogeneous catalysts [14]. Considering iron can be fixed into some adsorbents such as sepiolite [15], the regeneration of this silicate-based adsorbent using the EF process can be a good alternative to the aforementioned traditional adsorbent regeneration techniques.

The present article is novel in terms of the effective utilization of iron-loaded sepiolite as an inexpensive catalytic adsorbent with significant uptake on fixed-bed and batch adsorption studies for the targeted IL, 1-butylpyridinium chloride ([bpy] Cl,), whose elimination has not been studied before. Indeed, the strong interaction of IL–sepiolite makes desorption difficult. Thus, the exhausted adsorbent would be directly treated by the EF process. The strategy proposed is the combination of an adsorption process (that uses as sorbent one material which can act also as catalyst) with an EF process, with the idea of achieving the total elimination of the contaminant from the environment. Thus, the addition of iron to the sepiolite makes possible the use of this silicate-based material as a heterogeneous catalyst for the EF process and its application without the generation of iron sludge. After optimizing the operational variables, iron-loaded sepiolite was efficiently recovered through the EF process. Later on, the utilization of the regenerated adsorbent was assessed.

## 2. Results and Discussion

### 2.1. Adsorbents Screening

To choose a favorable adsorbent, many factors should be put into consideration. Among them, the adsorption capacity. Therefore, in this study, six low-cost adsorbents were examined using a 10 mg $L^{-1}$ [bpy] Cl polluted effluent (Section 3.2.1), aiming to find the best adsorbent for the [bpy] Cl elimination. Depicted in Table 1 are the obtained removals (R (%)) after the adsorption process. Under the conditions previously described, perlite presented the lowest adsorption removal among the tested adsorbents with an almost negligible value. Charred attapulgite and zeolite 13× presented intermediate removal values (61.18% and 57.85%, respectively). This fact may be due to ion exchange mechanisms [16]. Attapulgite, NaY zeolite and sepiolite seem to be very good adsorbents for [bpy] Cl as they showed very high removal efficiencies (above 94%). To select the best alternative, the maximum uptakes (Q) of these adsorbents were calculated after increasing the initial [bpy] Cl concentration from 10 to 1000 mg $L^{-1}$. Attapulgite and NaY zeolite maximum uptakes were around 7 mg $g^{-1}$, whereas that of sepiolite reached 32.02 mg $g^{-1}$. Thus, sepiolite would need to be replaced or regenerated much less often than the others when working in continuous mode, making this adsorbent a much better alternative.

**Table 1.** Screening of the selected adsorbents for the [bpy] Cl elimination.

| Adsorbent | Q (mg g$^{-1}$) | R (%) |
|---|---|---|
| Perlite beads | 0 | 0 |
| Charred attapulgite | 0.287 | 61.18 |
| NaY Zeolite | 0.485 | 95.98 |
| Zeolite 13X | 0.282 | 57.85 |
| Attapulgite | 0.484 | 94.78 |
| Sepiolite | 0.486 | 100 |

## 2.2. Desorption Tests

To regenerate the exhausted S-Fe, different desorbing agents were tested under the same conditions, and the obtained results are summarized in Table 2. Initially, desorption tests were conducted in neutral, acid and basic media [17]. Very low desorption efficiencies were shown which demonstrates that the principal mechanism implicated in the [bpy] Cl binding to the adsorbent was not ion-exchange. $H_2O$/acetonitrile (20/80) with $NH_4Cl$ (0.2 M) proved a high capacity to desorb a similar IL from natural adsorbents in our previous study [18]. Indeed, it was able to desorb 87.64% of 1-butyl, 2,3-dimethylimidazolium from attapulgite. Consequently, this eluent was also tested, as well as the decoupled agents (water and acetonitrile or an aqueous solution of $NH_4Cl$). Despite the improvement in the desorption efficiency with the combined alternative (Table 2), this desorbing agent is still not capable of removing the majority of [bpy] Cl from the adsorbent. Given these weak desorption results, a more effective regeneration method, such as the EF process, is needed to be used. Indeed, $Na_2SO_4$ desorption capacity was also assessed because it is used as an electrolyte on the EF test. The low desorption attained demonstrates that the degradation of [bpy] Cl might happen mainly in the adsorbent.

**Table 2.** Desorption levels (D%) of [bpy] Cl from sepiolite using different desorbing agents.

| Eluent | D% |
|---|---|
| $H_2SO_4$ (0.1 M) | 5.9 |
| NaOH (0.1 M) | 7.3 |
| HCl (0.1 M) | 10.7 |
| $Na_2SO_4$ (0.1 M) | 8.3 |
| $H_2O$ | 0 |
| $H_2O$/acetonitrile (20/80) with $NH_4Cl$ (0.2M) | 62.6 |
| $H_2O$/acetonitrile (20/80) | 7.2 |
| $NH_4Cl$ (0.2M) | 21.3 |

## 2.3. Catalytic Adsorbent Preparation and Characterization

The EF process has proven its efficiency in the regeneration of a variety of adsorbents in sludge medium [7,18,19]. Therefore, this process was chosen for regenerating the exhausted S-Fe. In the EF process, iron should be added to act as a catalyst. Thus, it is necessary to work at a pH around 3 to avoid iron precipitation and sustain the kinetics of $Fe^{2+}$ conversion to $Fe^{3+}$ [20]. However, in the sludge medium, fixing the pH around 3 is not always possible, especially when the adsorbent acidic buffering capacity is high. Indeed, in our previous study [19], it was necessary to add citric acid (complexing agent) to stabilize the iron and control the pH. Therefore, fixing the iron in the adsorbent can be a good alternative. Adsorbent characterization took place before and after the iron addition was done. Sepiolite and S-Fe have a conductivity of around 400 mS cm$^{-1}$, whereas their density is 1.68 g mL$^{-1}$. The pH$_{PZC}$ of both adsorbents is around 8.35. At this pH value, the surface has zero charge, while at pH values below and above the pH$_{PZC}$, it is positively and negatively charged, respectively. This pH$_{PZC}$ value is considered very high when compared to published data [21,22]. Acid and basic buffering capacities of both adsorbents were around 8.5 and 9.5, respectively. Those adsorbents seem

to have very high acid and basic buffering capacities when compared to kaolinite [23], which is also an efficient adsorbent. Pore size is a key parameter. Indeed, smaller sizes involve harder regeneration processes but higher adsorption rates [24]. The average pore size of sepiolite and S-Fe is around 109 Å, and their average pore volume is around 0.22 $cm^2 g^{-1}$. Those values are comparable to other silicate-based materials (illite and montmorillonite) previously studied for IL removal [7]. High surface area enhances the collision frequency between particles which implies better adsorption rates [25]. BET surface areas of sepiolite and S-Fe are 144.54 and 140.91 $m^2 g^{-1}$, respectively. Although these surface areas did not reach that of activated carbon, they are considered very high in comparison with those of other low-cost adsorbents such as raw chitosan and modified chitosan-pandan (1.079 and 1.31 $m^2 g^{-1}$, respectively) [26]. $N_2$ adsorption–desorption isotherms obtained for sepiolite and S-Fe are depicted in Supplementary Material (Figure S1). From this figure, it can be noticed that the adsorption isotherms are of type II regarding the classification of the International Union of Pure and Applied Chemistry (IUPAC) and have a hysteresis shape to a certain degree [22,27]. Thus, these adsorbents behave as porous media assembling macropores and mesopores with fewer micropores [25]. Moreover, there is no parallelism between the relative pressure axis and the saturation plateau, which indicates that the adsorbents have a large portion of mesopores [22]. The hysteresis loop belongs to type H3; therefore, slit-shaped pores are expected [28]. X-ray photoelectron spectroscopy (XPS) analysis of sepiolite happened before and after iron addition was done, and the obtained results are depicted in Figure S2 and Table S3. From these results, it can be noticed that the significant elements detected in both sepiolite and S-Fe samples were oxygen, magnesium, silicon and carbon. Similar results have been found by Wu and co-workers [29]. Moreover, some minor elements were also detected such as fluorine, iron and aluminum. From Table S3, it can be seen that there is no significant change in the elemental composition between sepiolite and S-Fe, while it can be noticed that in the S-Fe sample the iron amount was double that in sepiolite, which proves the successful iron attachment.

## 2.4. Kinetic and Isotherm Adsorption Studies

### 2.4.1. Kinetics

Firstly, [bpy] Cl adsorption kinetics have been studied by analysis of the adsorption of this IL within time into sepiolite and S-Fe. After that, intra-particle diffusion, pseudo-first order and pseudo-second order models have been evaluated to determine the model that fits the experimental data (Figure 1a,b).

The adsorption profiles of [bpy] Cl on sepiolite and S-Fe, depicted in Figure 1a,b, demonstrate the suitability of these adsorbents to be used for [bpy] Cl adsorption. It can be seen from these profiles that the adsorption has taken place quickly, reaching the apparent equilibrium in less than 30 min. Considering sepiolite and S-Fe's natural pH (8.67 > $pH_{PZC}$) and the fact that the [bpy]Cl organic part is positively charged, the good interaction between the adsorbent and the IL can be explained. The maximum uptake for sepiolite in 120 min was 31.64 mg $g^{-1}$, while S-Fe's maximum uptake was 22.85 mg $g^{-1}$. This decrease in the maximum adsorbent removal (around 28 %) may be caused by the repulsion between $Fe^{2+}$ fixed on the adsorbent and the organic part of [bpy] Cl. Other studies previously published [12,30] reported a decrease in the adsorbent performance due to the iron deposition. Moreover, it is important to highlight this adsorption decrease is in benefit of the subsequent EF regeneration using the catalytic adsorbent (S-Fe). Despite the decrease in its maximum uptake after iron deposition, this low-cost catalytic adsorbent has still a great adsorption capacity when it is compared with other adsorbents used for ILs uptake [7,18].

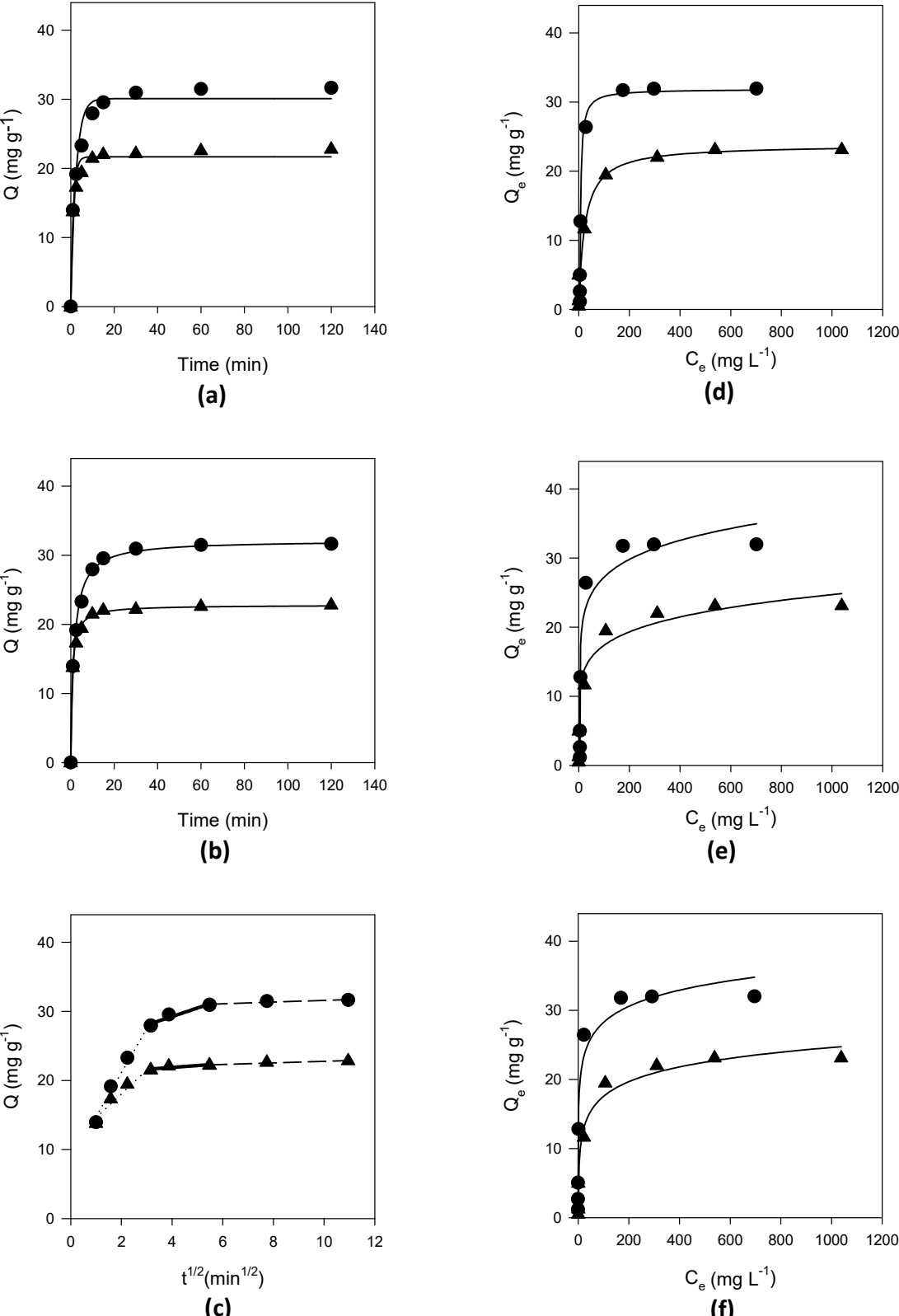

**Figure 1.** Adjustment of different models to the experimental data. Dots represent the plot of sepiolite data and triangles represent the plot of S-Fe data; (**a**) lines represent pseudo-first order model adjustments; (**b**) lines represent pseudo-second order model adjustments; (**c**) lines represent intra-particle diffusion adjustments; (**d**) lines represent Langmuir model adjustments; (**e**) lines represent Freundlich model adjustments; (**f**) lines represent Temkin model adjustments.

The parameters value as well as the $R^2$ and the standard error of the estimate (SEE) of the used kinetic models are summarized in Table 3. It can be noticed from this table the inadequacy of the pseudo-first order kinetic model for both adsorbents. Similar results were previously reported during the adsorption of 1-butyl, 2,3-dimethylimidazolium chloride onto illite and montmorillonite [7] as well as during the adsorption of 1-butyl, 2,3-dimethylimidazolium chloride onto attapulgite [18]. However, high $R^2$ values (0.995 and 0.998 for sepiolite and S-Fe, respectively) and low SEE values (0.773 and 0.348 for sepiolite and S-Fe, respectively) were obtained by the pseudo-second order model, showing that kinetic adjustment was successfully fitted to this model. Moreover, at equilibrium, the maximum uptakes calculated using this model were in concordance with the maximum experimental uptakes. The pseudo-second order kinetic model applicability suggests that chemisorption is among the mechanisms that have been implicated in the adsorption process [31]. To further understand the adsorption mechanism, the intra-particle diffusion model has been used. The poor adjustment of this latter model to the experimental data is given by the low $R^2$ reached (Table 3 and Figure S4). However, this model defines three linear regions (Figure 1c) with $R^2 > 0.87$ (Figure S4). Consequently, two or more controlling steps have been implicated in the [bpy] Cl adsorption onto both adsorbents [32].

**Table 3.** Parameters of kinetic and isotherm models used for fitting [bpy] Cl adsorption into sepiolite and S-Fe. Experimental conditions: adsorbent = 1 g; $V_s$ = 50 mL; 150 rpm; T = 25 °C; the contact time was fixed at 2 h for kinetics experiments and 16 h for the isotherms experiments.

| Adsorbent | Kinetic Models | | | Isotherms Models | | |
|---|---|---|---|---|---|---|
| | Model | Parameters | Values | Model | Parameters | Values |
| Sepiolite | Pseudo-first order model | $k_1$ (min$^{-1}$) $Q_e$ (mg g$^{-1}$) $R^2$ SEE | 0.415 30.095 0.965 2.113 | Freundlich | $K_F$ (mg$^{1-1/n}$ L$^{1/n}$ g$^{-1}$) n $R^2$ SEE | 24.397 22.694 0.834 1.048 |
| | Pseudo-second order model | $k_2$ (g mg$^{-1}$ min$^{-1}$) $Q_e$ (mg g$^{-1}$) $R^2$ SEE | 0.02 32.25 0.995 0.773 | Langmuir | $K_L$ (L mg$^{-1}$) $Q_{max}$ (mg g$^{-1}$) $R^2$ SEE | 0.298 31.923 0.973 2.496 |
| | Intra-particle diffusion model | $K_{id}$ (mg g$^{-1}$ min$^{-1}$) $I_d$ (mg g$^{-1}$) $R^2$ SEE | 1.509 19.189 0.612 4.417 | Temkin | $k_T$ (L mg$^{-1}$) $b_T$ (J mol$^{-1}$) $R^2$ SEE | 41.979 723.143 0.956 3.169 |
| S-Fe | Pseudo-first order model | $k_1$ (min$^{-1}$) $Q_e$ (mg g$^{-1}$) $R^2$ SEE | 0.824 21.768 0.97 1.272 | Freundlich | $K_F$ (mg$^{1-1/n}$ L$^{1/n}$ g$^{-1}$) n $R^2$ SEE | 8.586 6.536 0.942 2.584 |
| | Pseudo-second order model | $k_2$ (g mg$^{-1}$ min$^{-1}$) $Q_e$ (mg g$^{-1}$) $R^2$ SEE | 0.062 22.72 0.998 0.348 | Langmuir | $K_L$ (L mg$^{-1}$) $Q_{max}$ (mg g$^{-1}$) $R^2$ SEE | 0.042 23.71 0.962 2.096 |
| | Intra-particle diffusion model | $K_{id}$ (mg g$^{-1}$ min$^{-1}$) $I_d$ (mg g$^{-1}$) $R^2$ SEE | 0.695 17.106 0.544 2.344 | Temkin | $k_T$ (L mg$^{-1}$) $b_T$ (J mol$^{-1}$) $R^2$ SEE | 3.011 805.97 0.949 2.408 |

## 2.4.2. Isotherms

To evaluate the adsorbent capacity of the selected adsorbent more accurately than in the preliminary tests, the inlet effluent should be increased (preliminary tests were done with 10 mg L$^{-1}$). Thus, equilibrium isotherms for sepiolite and S-Fe adsorbents were evaluated at 25 °C with [bpy] Cl concentrations which ranged from 25 to 1500 mg L$^{-1}$ (Figure 1d–f). The obtained maximum uptakes after 16 h were 32.02 and 23.08 mg g$^{-1}$ for sepiolite and S-Fe, respectively. This high adsorption capacity as well as their low-cost make these adsorbents attractive alternatives to be employed instead of other reported adsorbents such as activated carbon.

Table 3 shows that the Langmuir model is suitable to adjust the experimental data. Indeed, it presented the highest $R^2$ and the lowest SEE values. In conclusion, the adsorbent has a homogeneous

monolayer surface. Besides, the adsorption occurred at specific sites and no interaction took place between the adsorbates [33]. Furthermore, the Langmuir maximum uptakes (31.923 and 23.71 mg g$^{-1}$ for sepiolite and S-Fe, respectively) were similar to the experimental data (32.02 and 23.08 mg g$^{-1}$ for sepiolite and S-Fe, respectively). The Langmuir isotherm applicability can be also expressed by the dimensionless separation factor $R_L$ (Equation (1)) [34]:

$$R_L = \frac{1}{1 + K_L \cdot C_a} \tag{1}$$

where $C_a$ is the pollutant concentration (mg L$^{-1}$) and $K_L$ is the Langmuir constant.

The $R_L$ value indicates the adsorption nature. Indeed, the adsorption can be unfavorable ($R_L > 1$), linear ($R_L = 1$), favorable ($0 < R_L < 1$) or irreversible ($R_L = 0$) [31]. The calculated values of $R_L$ were 0.03 and 0.15 for sepiolite and S-Fe, respectively. Consequently, under the studied conditions, the adsorption process is favorable.

### 2.5. Fixed-Bed Adsorption and Breakthrough Curve

For industrial applications, it is necessary to evaluate the utilization of this adsorbent at a bigger scale. Then, it is essential to have information about the pollutant concentration's dynamic behavior over time. Thus, a fixed-bed column containing the adsorbent was used to insert the pollutant solution continuously, so the outlet concentration was regularly measured. A breakthrough curve (BTC), which is the plot of C/C$_0$ with the inlet volume or time, was then obtained [35]. To obtain the BTC for [bpy] Cl, 200 mg L$^{-1}$ of [bpy] Cl solution was inserted in continuous mode into a fixed-bed column containing S-Fe, and the concentration of IL was measured over time. The fixed-bed column was filled up with around 30 g of S-Fe, which corresponds to 68.34 mL working volume in the column. From Section 2.4.1., it can be concluded that the apparent adsorption equilibrium was attained after approximately 30 min (Figure 1a,b). However, to be on the safe side and to avoid a too high flow which may damage the column and/or create preferential flow paths, 45 min was selected as the hydraulic retention time. According to the column set-up, this hydraulic retention time corresponds to a 1.52 mL min$^{-1}$ inlet flow. Iron was measured at the outlet of the column and on the adsorbent that had come out from the column after the adsorption process. No iron was detected in the outlet and the iron content in the adsorbent was the same as it was initially (11.55 mg g$^{-1}$), thus it was properly attached to the adsorbent. As it is shown in Figure 2, the BTC was characterized by its "S" shape. The breakpoint is obtained when fixing the concentration value at $C_b = 0.2\,C_0$, which was attained after 30 h when the treated volume was 2.74 L. The exhaustion point corresponds to a concentration of saturation ($C_s = 0.9\,C_0$). This point corresponds to more than 50 h treatment time and 4.56 L treated volume. Considering that the initial [bpy] Cl concentration was quite high, it is reasonable to think this process would be suitable for future uses for the remediation of high volumes of lower IL concentrations.

For the analysis of the experimental breakthrough curves, three standard models, Adams–Bohart (Equation (2)), Thomas (Equation (3)) and Yoon–Nelson (Equation (4)), were used.

$$\frac{C}{C_0} = \exp\left(k_A \cdot C_0 \cdot t - \frac{k_A \cdot N_0 \cdot H}{u}\right) \tag{2}$$

$$\frac{C}{C_0} = \frac{1}{1 + \exp\left(\frac{k_T \cdot q_0 \cdot M}{f} - k_T \cdot C_0 \cdot t\right)} \tag{3}$$

$$\frac{C}{C_0} = \frac{\exp(k_Y \cdot t - k_Y \cdot \tau)}{1 + \exp(k_Y \cdot t - k_Y \cdot \tau)} \tag{4}$$

where $k_A$ (L mg$^{-1}$ min$^{-1}$) is the rate constant of Adams–Bohart, $k_T$ (L mg$^{-1}$ min$^{-1}$) is the rate constant of Thomas, $k_Y$ (min$^{-1}$) is the rate constant of Yoon–Nelson, $u$ (cm min$^{-1}$) is the superficial velocity, $H$ (cm) is the length of the column, $q_0$ (mg g$^{-1}$) is the adsorbent capacity, $N_0$ (mg L$^{-1}$) is the concentration of

saturation, f (mL min$^{-1}$) is the flow rate, $\tau$ (min) is the time needed for 50% adsorbate breakthrough, and M (mg) is the adsorbent amount in the column.

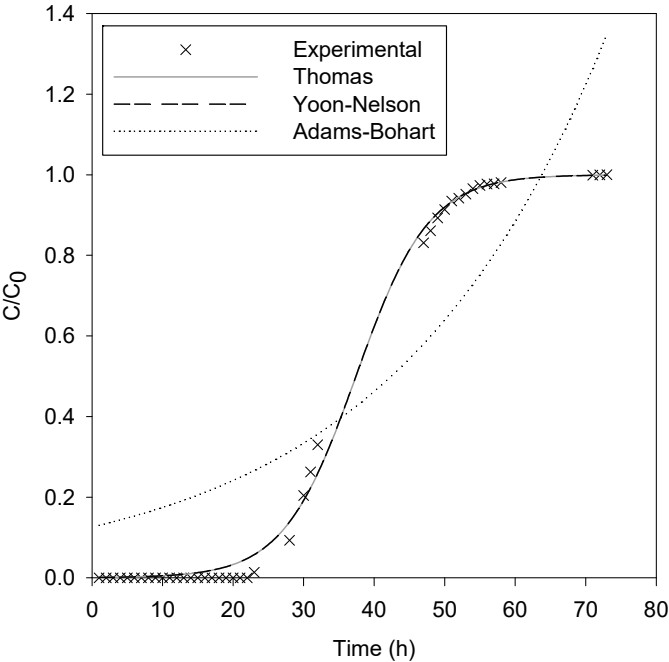

**Figure 2.** Experimental data obtained in the breakthrough study of the [bpy] Cl-S-Fe system and fitting of the experimental breakthrough data to mathematical models. Operational conditions: fresh [bpy] Cl solution concentration = 200 mg L$^{-1}$; column height = 14 cm; flow rate = 1.52 mL min$^{-1}$.

Table 4 shows a good fit for breakthrough data by the Yoon–Nelson and Thomas models ($R^2$ = 0.998 and SEE = 0.02). By contrast, a bad fit was attained by the Adams–Bohart model ($R^2$ = 0.805 and SEE = 0.188). On the one hand, the good adjustment of the Yoon–Nelson kinetic model means that the decrease rate of the probability of adsorption is related to the pollutant adsorption and breakthrough probability [36]. On the other hand, Thomas adjustment affirms the previously obtained data, as this model anticipates breakthrough behavior assuming pseudo-second order kinetics and a Langmuir isotherm [37]. This latter breakthrough kinetic model is applicable when internal and external diffusions are not limiting steps [36]. The calculated parameters of the Yoon–Nelson and Thomas kinetic models are in concordance with the experimental data. Indeed, the adsorption capacity of S-Fe calculated using the Thomas kinetic model is 22.77 mg g$^{-1}$, which is close to that reported in Section 2.4.2. (23.08 mg g$^{-1}$). Likewise, in the Yoon–Nelson kinetic model, $\tau$ was 37.43 h (2245.56 min), which is comparable to the 38 h experimentally determined.

After adsorption, the fixed-bed column was divided into five sections and the pollutant distribution into these sections was examined (Figure S5) (where Section 1 is the closest to the inlet opening of the column). As it can be noticed from Figure S5, the minimum uptakes have been found in the latest sections of the column, close to the outlet. These sections did not reach the maximum uptake even if fresh [bpy] Cl solution was passed through the column. Lei and Zhou [38] justified this fact by the presence of preferential flow paths. To enhance the [bpy] Cl loading, changing the flow direction could be a solution. The average [bpy] Cl concentration after the column adsorption was around 20.51 mg g$^{-1}$. After that, all of the [bpy] Cl loaded adsorbent was mixed and total organic carbon (TOC) was directly measured in the solid matrix. The obtained experimental TOC was around 20.8 mg g$^{-1}$.

**Table 4.** Breakthrough curve models' parameters for the [bpy] Cl-S-Fe system. Operational conditions: fresh [bpy] Cl solution concentration = 200 mg $L^{-1}$; column height = 14 cm; flow rate = 1.52 mL $min^{-1}$.

| Model | Parameters | Values |
|---|---|---|
| Thomas | $q_0$ (mg $g^{-1}$) | 22.77 |
| | $k_T$ (L $mg^{-1}$ $min^{-1}$) | $1.617 \cdot 10^{-5}$ |
| | $R^2$ | 0.998 |
| | SEE | 0.02 |
| Yoon–Nelson | $k_Y$ ($min^{-1}$) | $3.234 \cdot 10^{-3}$ |
| | $\tau$ (min) | 2245.56 |
| | $R^2$ | 0.998 |
| | SEE | 0.02 |
| Adams–Bohart | $K_A$ (L $mg^{-1}$ $min^{-1}$) | $162.348 \cdot 10^{-6}$ |
| | $N_0$ (mg $L^{-1}$) | 899.935 |
| | $R^2$ | 0.805 |
| | SEE | 0.188 |

*2.6. AOPs Adsorbent Regeneration Tests*

2.6.1. AOPs Preliminary Assays

Fenton regeneration trials were conducted by adding 10 mL of $H_2O_2$ aqueous solution at different concentrations (3, 30 and 300 mM) to 1 g of exhausted adsorbent. A preliminary Fenton test was executed by adding 300 mM $H_2O_2$ solution to sepiolite polluted with [bpy] Cl. The inherent metals on the sepiolite structure were not sufficient to carry out the Fenton's reaction (Equation (5)) as no mineralization was noticed.

$$Fe^{2+} + H_2O_2 \rightarrow Fe^{3+} + HO^\bullet + OH^- \tag{5}$$

After that, Fenton experiments took place using exhausted S-Fe. The iron amount into the S-Fe was around 11.5 mg $g^{-1}$ which makes 20 mM of iron go into the reactor. This quantity was stable throughout the process as no iron was detected during the process. The $H_2O_2$/iron ratio range from 0.15 to 15 was tested, which is within the range that other authors have studied [39]. Table 5 resumes the obtained results. From these experiences, it can be noticed that TOC was reduced which proves that S-Fe works well as a catalytic adsorbent. Furthermore, $H_2O_2$ concentration affected the Fenton process performance (Table 5), but only a maximum of 26.47% mineralization was achieved after 24 h treatment time. These results are in concordance with Banic et al. [40], who degraded 27% of the ionic liquid 3-methylimidazolium chloride under a homogeneous Fenton process which needed to be coupled to photocatalysis for having an acceptable degradation rate (80%). Thus, the Fenton process itself seems to be not efficient for ionic liquid degradation and adsorbent regeneration even with the addition of high $H_2O_2$ concentrations. As a result, a more effective process should be used.

AO tests were carried out over 9 h, in the same EF reactor, and under three current intensities (100, 300 and 500 mA). For that, [bpy] Cl was adsorbed into sepiolite to have an initial TOC of [bpy] Cl of 20.8 mg $g^{-1}$. Then, 10 g of this polluted adsorbent was added to 200 mL of 0.1 M $Na_2SO_4$ solution to be treated. Taking into consideration the ionic nature of [bpy] Cl, the concentration of it in the solution during the AO process was monitored and no IL was detected, demonstrating either the IL is being degraded directly into the sepiolite or if desorption is happening, it is the rate limiting step when coupled to the actual AO degradation. The obtained results shown in Table 5 show that rising the current intensity from 100 to 300 mA favored the oxidation of [bpy] Cl due to the greater HO· generation according to Equation (6) [41]. Therefore, the TOC removal increased from 13.36% to 31.01%. However, a further increase in the current until 500 mA did not produce significant improvement in the oxidation process. Indeed, the TOC removal was around 34.42%. This fact can be justified by the

promotion of parasitic reactions at great current intensities, primarily oxygen reduction [42], hydrogen production at the cathode and oxygen generation at the anode [43].

$$H_2O \rightarrow HO^{\bullet}_{ads} + H^+ + e^- \tag{6}$$

**Table 5.** Experimental conditions of advanced oxidation processes (AOPs) used for S-Fe regeneration and consequent TOC removal.

| Trial Number | AOP Treatment | [H$_2$O$_2$] (mM) | I (mA) | pH | [Na$_2$SO$_4$] (M) | Treatment Time (h) | TOC Removal (%) |
|---|---|---|---|---|---|---|---|
| 1 | Fenton | 3 | - | 3 | - | 24 | 4.58 |
| 2 | Fenton | 30 | - | 3 | - | 24 | 19.01 |
| 3 | Fenton | 300 | - | 3 | - | 24 | 26.47 |
| 4 | AO | - | 100 | 8.6 | 0.1 | 9 | 13.36 |
| 5 | AO | - | 300 | 8.6 | 0.1 | 9 | 31.01 |
| 6 | AO | - | 500 | 8.6 | 0.1 | 9 | 34.42 |
| 7 | AO | - | 300 | 3 | 0.1 | 9 | 31.65 |
| 8 | EF | - | 300 | 8.6 | 0.1 | 9 | 43.23 |
| 9 | EF | - | 300 | 3 | 0.1 | 9 | 56.98 |
| 10 | EF | - | 300 | 3 | 0.2 | 9 | 82.79 |
| 11 | EF | - | 300 | 3 | 0.3 | 9 | 93.70 |

### 2.6.2. EF Treatment

An EF regeneration experiment was conducted under a current intensity of 300 mA by adding 10 g of S-Fe to 200 mL of 0.1 M Na$_2$SO$_4$ solution. During the EF process, no iron was detected in the bulk solution, demonstrating the stability of the S-Fe catalytic adsorbent. As can be seen in Table 5, the TOC removal was around 57%, defeating the AO performance due to the generation of radicals throughout the Fenton process [13] and also enhancing the Fenton performance because of the electric field application. This synergistic effect has been explained by the fact Fe$^{2+}$ and H$_2$O$_2$ are, respectively, regenerated (from Fe$^{3+}$) and generated (from the provided O$_2$). Thus, this allows the process to continue over time [13].

Nevertheless, to ensure the pH modification is not responsible for this amelioration, an AO test was done at pH 3, where slight differences were detected (Table 5). Similar results about the minor effect of initial pH on the oxidation of organic pollutants using BDD oxidation have been previously reported [44,45]. Moreover, an initial EF test at neutral pH was carried out, improving slightly the mineralization performance when compared to AO. Nevertheless, working at the optimal Fenton pH (around 3) increased the mineralization from 43.23 to 56.98 because of the Fe$^{2+}$ being more available and the Fenton process favored [13]. The performance of the system can be ameliorated by increasing the electrolyte concentration, as the slurry reactor has a high medium resistance leading to improper energy distribution. To test the effect of the electrolyte concentration on the process performance, EF assays were done by increasing the Na$_2$SO$_4$ concentration two and three times. The obtained results (Table 5) show an increase in the TOC removal when increasing the electrolyte concentration, reaching 93.7% TOC removal with a Na$_2$SO$_4$ concentration of 0.3 M. Other authors have also reported the amelioration of the EF process performance when reducing the media resistance [46]. The higher amelioration, in this case, may be caused by the generation of additional oxidants following Equations (7) and (8) [47].

$$SO_4^{2-} \rightarrow SO_4^{-\bullet} + e^- \tag{7}$$

$$SO_4^{-\bullet} + SO_4^{-\bullet} \rightarrow S_2O_8^{2-} \tag{8}$$



### 2.7. TOC Measurement and Carboxylic Acids Determination

TOC in the liquid as well as in the solid matrix has been measured over 24 h of EF treatment under 300 mA and with 0.3 M $Na_2SO_4$ solution. The obtained results are depicted in Figure 3. From Figure 3a, it can be noticed that total adsorbent remediation was attained after 10 h of EF treatment. These results defeat previously reported EF regeneration processes due to the appropriate selection of the adsorbent and process optimization. Indeed, even our previous work [7] was ameliorated where a slurry system was used for the degradation of the IL 1-butyl, 2,3-dymethyl imidazolium chloride. Without having optimized the process, the IL degradation achieved after 6 h was 75% and 33% when using, respectively, Illite and Montmorillonite because on the former, electro-desorption occurred. In the current study, even mineralization is higher, as after 6 h, 80% of TOC abatement was attained (Figure 3a). Another actual example is Acevedo-García et al. [48], who attained 70% of mineralization when regenerating polluted biochar with either sulfamethoxazole or methyl paraben. Exhausted S-Fe regeneration induced the generation of by-products in the bulk solution. This fact is demonstrated in Figure 3b by the increase in the dissolved TOC until the first six treatment hours, and its reduction after that, reaching a value of 70.3 mg $L^{-1}$ after 24 h. Some detected aliphatic carboxylic acids were responsible for 55.8% of the remaining TOC (Figure 3b).

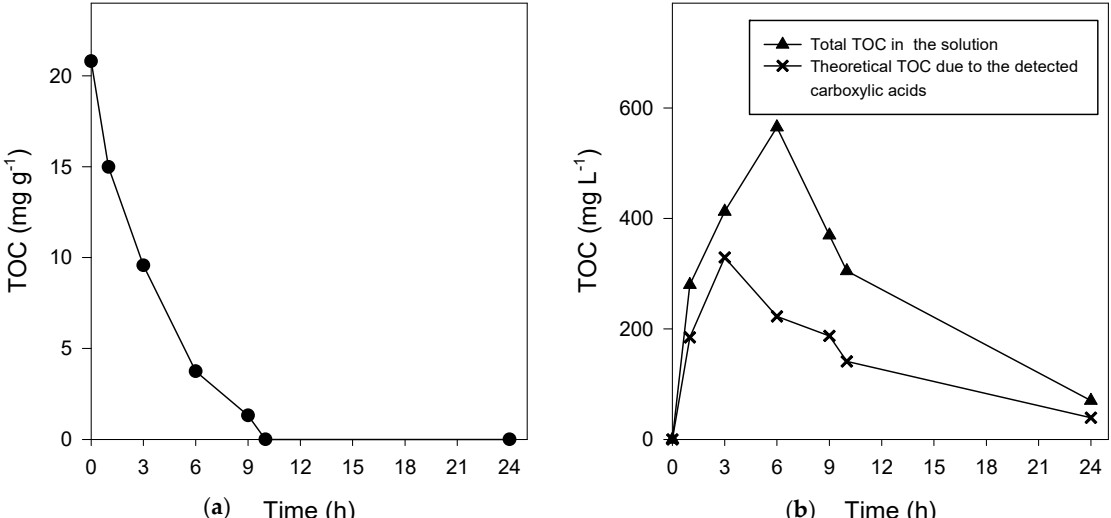

**Figure 3.** TOC measurement in (**a**) the adsorbent and in (**b**) the solution during treatment under optimal conditions.

Indeed, seven carboxylic acids (oxalic, malonic, succinic, glycolic, formic, acetic and oxamic acids) have been quantified and identified (Figure 4). Figure 4 shows that the carboxylic acids concentration profiles presented accumulation–destruction stages. Similar behavior for different organic compounds treatment has been previously observed [49,50]. Several carboxylic acids were generated during the EF treatment; however, high oxalic and acetic acids concentrations were found. On the other hand, succinic and glycolic acids were detected in small quantities, then they were degraded at short treatment times (6 and 9 h, respectively). According to Pimentel and co-workers [51], succinic acid is degraded into formic acid and then into $CO_2$. Due to their low reactivity with ·OH, oxalic and oxamic acids remained in the solution even after 24 h treatment. Similar results have been previously reported [52,53].

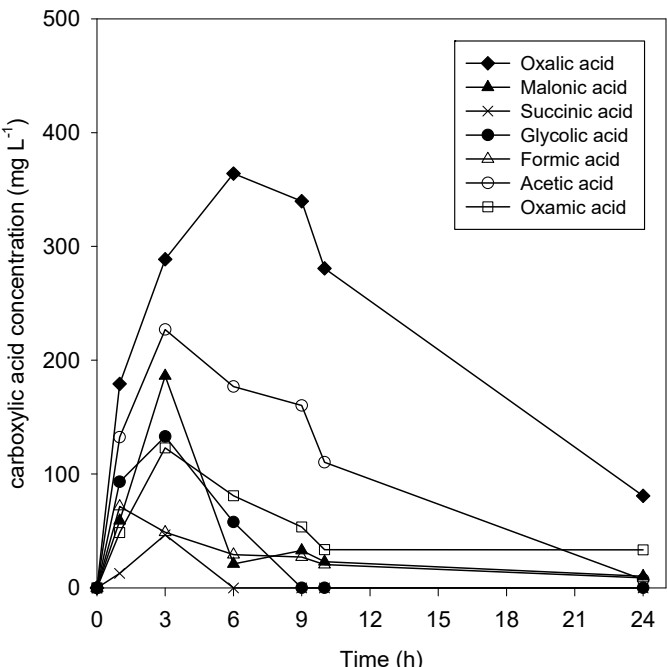

**Figure 4.** Concentration–time profile of carboxylic acids formed during the EF regeneration process.

## 2.8. Adsorbent Stability and Reuse

S-Fe stability was examined during four adsorption–regeneration cycles using XPS analysis and the obtained results are shown in Figure S6 and Table S6. Initially, it can be noticed that the exhausted S-Fe spectrum (spectrum B) has changed in comparison with that of S-Fe (spectrum A), as new elements, chlorine and nitrogen, were detected. Moreover, from Table S6, it can be noticed that the carbon atomic percentage has slightly increased. These modifications can be caused by the [bpy] Cl ($C_9H_{14}ClN$) adsorption on the S-Fe surface. Then, 10 g of exhausted S-Fe adsorbent was regenerated by the EF process over 9h using 200 mL of 0.3 M $Na_2SO_4$ solution and under an applied current of 300 mA. After each EF treatment, the obtained regenerated adsorbent was isolated, by filtration, from the reaction solution and then oven-dried at 40 °C for 24 h to be used in the next adsorption process. After adsorption, the adsorbent was dried again to be treated by EF. Figure S6 and Table S6 show that there is no big change in the elemental composition of S-Fe after three successive EF oxidations. However, they showed the appearance of a new element (sodium) with a progressive increase in its atomic percentage from the first to the third oxidation. This can be explained by the use of $Na_2SO_4$ as an electrolyte during the EF process. The atomic percentages of nitrogen, chlorine and carbon, indicative elements of [bpy] Cl presence, decrease strongly after each EF process, which proves the efficiency of the EF treatment. Figure 5 shows the [bpy] Cl uptakes into S-Fe during four adsorption–EF cycles. From this figure, it can be observed that the maximum adsorbent uptake has not been reduced even after four cycles. Moreover, in all cycles, the kinetic behavior of the IL adsorption on S-Fe was similar (Table 6). This can be explained by the stability of the adsorbent proved by the XPS analysis and by the fact that the adsorbent surface area is still almost the same during the four regeneration cycles (Table 6). It can also be explicated by the almost total recovery of the exhausted S-Fe adsorbent after each EF treatment. These results defeat previous EF regeneration processes (for kaolinite polluted with rodhamine B) where slight adsorption worsening was detected after the third regeneration cycle [54].

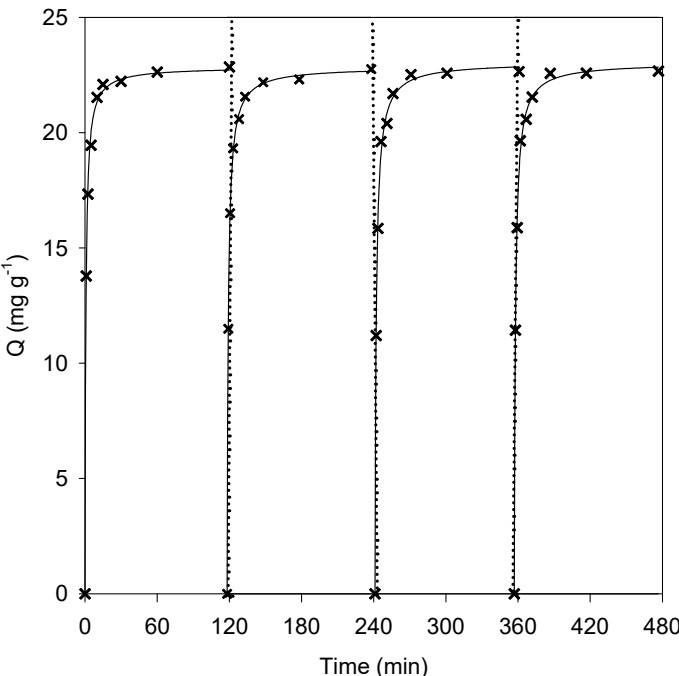

**Figure 5.** Uptakes of [bpy] Cl onto S-Fe during four adsorption–regeneration cycles. Full lines are the pseudo-second order kinetic fit.

**Table 6.** Pseudo-second order kinetic fit of [bpy] Cl adsorption onto S-Fe, regeneration efficiency and surface area during four adsorption–regeneration cycles.

| Cycle Number | Pseudo-Second Order Kinetic | | $n_r$ (%) | Surface Area ($m^2$ $g^{-1}$) |
|---|---|---|---|---|
| | Parameters | Values | | |
| 1 | $k_2$ (g $mg^{-1}$ $min^{-1}$) | 0.062 | 100 | 144.54 |
| | $Q_e$ (mg $g^{-1}$) | 22.72 | | |
| | $R^2$ | 0.998 | | |
| | SEE | 0.348 | | |
| 2 | $k_2$ (g $mg^{-1}$ $min^{-1}$) | 0.045 | 100 | 126.6 |
| | $Q_e$ (mg $g^{-1}$) | 22.72 | | |
| | $R^2$ | 0.999 | | |
| | SEE | 0.151 | | |
| 3 | $k_2$ (g $mg^{-1}$ $min^{-1}$) | 0.04 | 102.33 | 153.04 |
| | $Q_e$ (mg $g^{-1}$) | 23.25 | | |
| | $R^2$ | 0.998 | | |
| | SEE | 0.338 | | |
| 4 | $k_2$ (g $mg^{-1}$ $min^{-1}$) | 0.041 | 102.33 | 141.53 |
| | $Q_e$ (mg $g^{-1}$) | 23.25 | | |
| | $R^2$ | 0.998 | | |
| | SEE | 0.319 | | |

*2.9. Future Uses*

Considering the reported results and bearing in mind more studies need to be done, the real usage of this process can be spotted. Indeed, the 10 h EF regeneration treatment resulted in an energy consumption of around 1.33 kWh $kg^{-1}$. This value seems to be quite low in comparison with that found by Zhou and Lei (2.78 kWh $kg^{-1}$) during p-nitrophenol-loaded activated carbon electrochemical regeneration [55]. This consumption makes the process cost 0.000532 USD $L^{-1}$ (Table S7), which would be even less if moving from laboratory scale to industrial scale. This price is more encouraging than that

found by Gaied et al. [56], who found 0.02038 USD L$^{-1}$ (17.331 EUR m$^3$) for 3 h of the electro-Fenton process for the tertiary treatment of real wastewater. Considering ILs are prone to be at very low concentrations on effluents and considering the high uptake of S-Fe, quite small columns would be necessary. The regeneration step can be done consequently on small reactors, although the usage of bigger reactors for regenerating several batches could be deliberated, but in this case, the bigger working volume, the wider distance between electrodes and so on should be considered and optimized consequently. Another advantage of this process is the fact that the electrolyte solution could be reused, as well as the adsorbent as it was demonstrated in Section 2.8, which leads to a significant decrease in the operational costs.

## 3. Materials and Methods

### 3.1. Reagents

[bpy] Cl was bought from Iolitec (Heilbronn, Germany). Attapulgite and charred attapulgite (Tolsa, Madrid, Spain), as well as sepiolite (Distrilema químicos, S.L., Bamio, Galicia, Spain), zeolite 13X (Xiamen Zhongzhao Imp. & Exp. Co., LTD, Sha Xiamen, China), NaY zeolite (Zeolyst International, Conshohocken, PA, USA) and perlite beads (Sigma Aldrich, Madrid, Spain), were tested as adsorbents without further purification. All the used chemicals (NaOH, HCl, H$_2$SO$_4$, FeSO$_4$,7H$_2$O, etc.) had a purity ≥ 98% and have been purchased from Sigma Aldrich (Madrid, Spain).

### 3.2. Experimental Set-up

#### 3.2.1. Adsorbents Screening

Six adsorbents have been examined under the same adsorption conditions to evaluate their adsorptive capacity: 1 g of adsorbent was mixed with 50 mL of [bpy] Cl aqueous solution (10 mg L$^{-1}$). This mixture has been sustained at 150 rpm and 25 °C for 16 h using a shaker (MaxQ 8000, Thermo Scientific, Waltham, MA, United States). After that, the supernatant has been isolated from the adsorbent by filtration and its [bpy] Cl concentration was analyzed following the analytical procedure described below. The presented results are the average values of assays performed in duplicate.

The adsorptive capacity Q (mg g$^{-1}$) and the removal efficiency R (%) have been determined as follows (Equations (9) and (10)).

$$Q = \frac{(C_0 - C_t) \cdot V_s}{W} \tag{9}$$

$$R = \frac{(C_0 - C_t) \cdot 100}{C_0} \tag{10}$$

where W (g) is the mass of the adsorbent, V$_s$ (L) is the volume of the solution, C$_0$ (mg L$^{-1}$) is the initial [bpy] Cl concentration, and C$_t$ (mg L$^{-1}$) is the [bpy] Cl concentration at a precise time.

#### 3.2.2. Desorption

Firstly, 1 g of [bpy] Cl-loaded sepiolite was mixed with 50 mL of the desorbing solution. After that, the mixture was placed in an orbital shaker (MaxQ8000, Thermo Scientific) and kept overnight at 150 rpm and 25 °C to ensure maximum desorption. The tested desorbing solutions were: H$_2$O, HCl (0.1 M), NaOH (0.1 M), H$_2$SO$_4$ (0.1 M), Na$_2$SO$_4$ (0.1 M), NH$_4$Cl (0.2 M), H$_2$O/acetonitrile (20/80) and H$_2$O/acetonitrile (20/80) with NH$_4$Cl (0.2 M).

The desorption efficiency D (%) was measured as follows:

$$D = \frac{\frac{C_t \cdot V_s}{W}}{Q} \cdot 100 \tag{11}$$

### 3.2.3. Catalytic Adsorbent Preparation

Sepiolite doped with iron (S-Fe) was prepared to be used, at the same time, as an adsorbent and heterogeneous catalyst for the subsequent AOPs regeneration experiments. Iron amendment into sepiolite was carried out following Cabrera-Codony and co-workers' previous research [12]. Briefly, 10 mL of iron solution (4 g L$^{-1}$ Fe), using FeSO$_4$·7H$_2$O as an iron source, was added to 2 g of adsorbent. The pH was fixed at 2.5 by adding H$_2$SO$_4$ and the suspension was agitated for 4 d. After that, NaOH was used to raise the pH to 4. After that, the adsorbent was extracted by filtration, diluted in 10 mL of deionized water, and the mixture was agitated for 2 h. Finally, the adsorbent was again filtered and dried at 105 °C for 24 h, obtaining the S-Fe catalytic adsorbent.

### 3.2.4. Kinetics and Isotherms

#### Kinetics

Kinetic experiments were conducted by mixing 50 mL of 1000 mg L$^{-1}$ [bpy] Cl solution with 1 g of adsorbent. The mixture was stirred in an orbital shaker (MaxQ8000, Thermo Scientific) at 150 rpm and 25 °C for 2 h; meanwhile, samples were taken periodically. Then, pseudo-first order (Equation (12)), pseudo-second order (Equation (13)) and intra-particle diffusion (Equation (12)) kinetic models were used to fit the obtained experimental data.

$$Q = Q_e \cdot \left(1 - e^{-k_1 t}\right) \tag{12}$$

$$\frac{1}{Q} = \frac{1}{k_2 \cdot Q_e^2} + \frac{t}{Q_e} \tag{13}$$

$$Q = k_{id} \cdot t^{0.5} + I_d \tag{14}$$

where $Q_e$ (mg g$^{-1}$) is the uptake at the equilibrium, $Q$ is the uptake at a given time t (min), while $k_{id}$, $k_1$ and $k_2$ are the kinetic constants for intra-particle diffusion, pseudo-first order and pseudo-second order kinetic model, respectively.

#### Isotherms

An amount of 1 g of adsorbent was mixed with 50 mL of differently concentrated effluents (from 25 until 1500 mg L$^{-1}$) for 16 h in 250 mL Erlenmeyer flasks, which were placed into a shaker at 150 rpm and 25 °C. The attained results can be fitted to different models depending on the interaction of [bpy] Cl–adsorbent. The Langmuir isotherm (Equation (15)) fits well with monolayer adsorption into a homogeneous surface. Freundlich (Equation (16)) is adjusted to multilayer adsorption into a heterogeneous surface. The Temkin isotherm (Equation (17)) represents adsorption in which interactions of adsorbate–adsorbent cause a linear diminution on the adsorption heat because of the adsorbent coverage [57].

$$Q_e = \frac{k_L \cdot C_e \cdot Q_{max}}{1 + k_L \cdot C_e} \tag{15}$$

$$Q_e = k_F C_e^{\frac{1}{n}} \tag{16}$$

$$Q_e = \frac{R \cdot T}{b_T} \cdot \ln k_T C_e \tag{17}$$

Both isotherm and kinetic adjustments were selected accordingly to the coefficient of multiple regression (R$^2$) which should be close to 1 and to the standard error of estimate (SEE) (Equation (18)), which should be as small as possible.

$$SEE = \sqrt{\frac{\Sigma \left(Q_{exp} - Q_{theo}\right)^2}{n - 2}} \tag{18}$$

where $Q_{exp}$ (mg g$^{-1}$) is the experimental uptake, $Q_{theo}$ (mg g$^{-1}$) is the calculated uptake, and n is the number of the measurement.

### 3.2.5. Fixed-Bed Adsorption

[bpy] Cl adsorption onto S-Fe was conducted in a continuous flow system utilizing a glass column (Figure 6a). The fixed-bed column has a 2.8 cm inner diameter, 22.5 cm of length and 86.2 mL of capacity. At the bottom and the top of the column, a nylon sponge was used to prevent S-Fe from being washed away. An amount of 30 g of S-Fe was added to the column, which covered around 14 cm adsorbent bed length. To maintain a uniform flow distribution, glass beads have been used to fill up the extremes of the column. After that, [bpy] Cl solution (200 mg L$^{-1}$) was fed to the fixed-bed column at a flow rate of 1.52 mL min$^{-1}$, using a peristaltic pump (Masterflex L/S easy-load 77202–60, Cole-Parmer, Vernon Hills, United States). Finally, samples at the outlet solution were taken each 1 h, filtered and then analyzed. All the experiments have been conducted without pH adjustment at 25 °C.

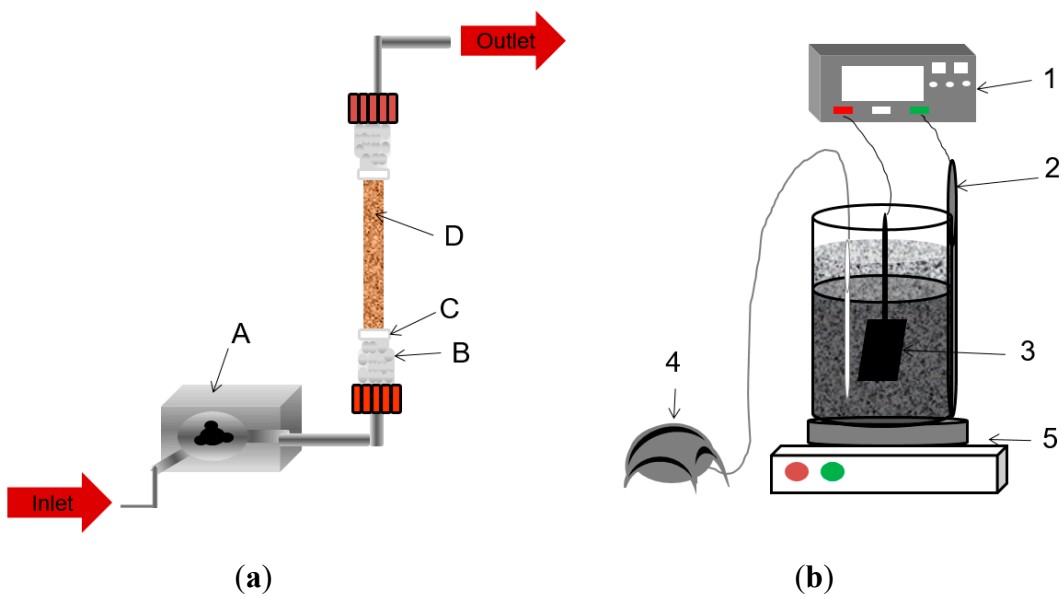

**Figure 6.** Experimental set-up (**a**) column reactor schema: A: peristaltic pump, B: glass beads, C: nylon sponge, D: S-Fe; (**b**) slurry reactor schema. 1: electric field supply, 2: cathode, 3: anode, 4: air pump, 5: magnetic stirrer.

### 3.2.6. AOPs S-Fe Regeneration Studies

The spent S-Fe was treated under Fenton and EF processes, thanks to the trapped iron into S-Fe which can act as a catalyst, to find a powerful strategy to reuse this S-Fe catalytic adsorbent.

#### Fenton Trials

Glass vials (12 mL) were used and 10 mL of $H_2O_2$ aqueous solution was added to 1 g of exhausted S-Fe. Various $H_2O_2$ concentrations were tested to favor the degradation (3, 30 and 300 mM). The experiments were conducted in a multi-purpose tube rotator (Mobile-Rod, J.P. Selecta, Abrera, Barcelona, Spain) at pH 3 for 24 h. Fenton assays were carried out in duplicate and the obtained results were the average values.

#### AO and EF Treatment

For AO and EF experiments, a cylindrical glass reactor was employed (Figure 6b). In the very middle of the cell, a Boron Doped Diamond (BDD) (Condias GmbH, Itzehoe, Germany) electrode with an area of 10.8 cm$^2$ was placed to be used as the anode. The cathode, which is a carbon felt

(Mersen, Saint Bonnet de Mure, France) with an area of 105 cm$^2$, surrounded the BDD and covered the inner glass of the cylindrical reactor. An amount of 200 mL of $Na_2SO_4$ aqueous solution was used as the working matrix, where 10 g of exhausted sepiolite or S-Fe was added in the, respectively, AO or EF processes. The current intensity was fixed at the selected value with the Blausonic power supply. Air was pumped at 1 L min$^{-1}$ to attain $H_2O_2$ generation (only on EF processes) and to help the S-Fe fluidization, which was enhanced by magnetic stirring (300 rpm) in both AO and EF processes.

### 3.3. Analytical Procedures

### 3.3.1. Adsorbent Characterization

#### Buffering Capacity

Buffering capacity was measured by a previously reported method [58]. Briefly, 4 g of adsorbent was mixed with 60 mL of distilled water. After 30 min of magnetic stirring, the mixture pH was determined. Then, 1 mL of NaOH or HCl 1 M was added, and after stirring for 30 min, the pH was measured again. This procedure was repeated several times aiming to determine the adsorbent response to pH modifications.

#### Point of Zero Charge (PZC)

PZC was evaluated following a reported procedure [59]. Briefly, different sets of 1 g of adsorbent with 50 mL of a 0.1 M $NaNO_3$ solution were prepared in Erlenmeyer flasks. The pH of each Erlenmeyer was adjusted at different values within the range (pH= 2 to 12) with either $HNO_3$ or NaOH. After having added the adsorbent to the solutions, they were shaken for 24 h (150 rpm at 25 °C). The samples were centrifuged, and the pH change in the aqueous fraction was measured.

#### Conductivity and pH

Conductivity was measured following the protocol UNE 77308:2001. Thus, 100 mL of distilled water was mixed with 20 g of the adsorbent for 30 min, and then the conductivity of the aqueous fraction was measured. pH was measured following the protocol UNE 77305:1999. For that, a given volume of adsorbent was mixed with five times the volume of distilled water. The mixture was stirred for 5 min and left to repose for 2 h. The pH was determined in the aqueous fraction.

#### Structure and Mineralogical Composition

The pore size, volume and BET surface area were measured using the Surface Area and Porosity Analyzer 2020 (Micromeritics, Norcross, GA, USA). Approximately 0.2 g of adsorbent was used for carrying out the measurement. Samples were degasified for 60 min at 90 °C and heated for 240 min at 100 °C. Both temperatures were reached at a rate of 10 °C min$^{-1}$ and the pressure was held at 0.13 atm. The adsorbent elemental and chemical composition was accomplished by XPS surface measurements (ESCALAB 250Xi, Thermo Fisher Scientific, Barcelona, Spain). These analyses were done by the external services of CACTI (University of Vigo).

### 3.3.2. Iron Content Evaluation

To ensure iron presence, S-Fe was acid-digested (1 g in 10 mL 7 M $HNO_3$ solution for 30 min at 120 °C and 1 atm in an autoclave). Then, the adsorbent was extracted by gravity filtration and raised to an exact known volume (in a volumetric flask). This solution as well as the bulk solution after both Fenton and EF processes were measured for iron determination. The quantification of the dissolved iron was done using the phenanthroline colorimetric method [60]. The samples were measured at 510 nm on the V−630 spectrophotometer (Jasco, Madrid, Spain).

### 3.3.3. High-Performance Liquid Chromatography (HPLC) Measurements

The results comparison was adequately carried out by the measurement of the [bpy] Cl and the carboxylic acids generated concentration in the solution using an HPLC (Agilent 1260, Las Rozas de Madrid, Madrid, Spain) coupled to a DAD detector. The [bpy] Cl concentration was measured using a gradient acetonitrile/aqueous phosphate buffer solution (5 mM $Na_2HPO_4$ and 7.5 mM $H_3PO_4$) which flowed at 1 mL min$^{-1}$ throughout the column Synergy 4μ Polar-RP 80A 4,6 × 150 mm (Phenomenex). The carboxylic acids were quantified using a 2.5 mM $H_2SO_4$ isocratic solution at 0.5 mL min$^{-1}$ in the column ROA-Organic Acid H$^+$ (8%), 300 × 7.8 mm (Rezex™), which was kept at 60 °C to enhance the carboxylic acids separation.

### 3.3.4. Total Organic Carbon (TOC) Measurements

The liquid samples' TOC measurement was done employing an Analytik Jena multi N/C 3100$^{®}$ coupled to an NDIR carbon detector (CACTI, Vigo) (Jena, Germany). To quantify the amount of [bpy] Cl onto S-Fe, TOC was also directly measured in the solid matrix using the elemental analyzer LECO CNS2000 (Labx, Midland, ON, Canada). After that, [bpy] Cl and TOC removal percentages were calculated following Equation (19).

$$R(\%) = \frac{(\alpha_0 - \alpha_t) \cdot 100}{\alpha_0} \tag{19}$$

where $\alpha_0$ and $\alpha_t$ are the studied parameter values before treatment and at a specific treatment time, respectively.

### 3.4. Energy Consumption

An evaluation of the energy parameters is essential to compare electrochemical-based processes. Thus, the energy consumption was determined as follows [42]:

$$\text{Energy consumption kWh/kg} = \frac{E \cdot I \cdot t}{W} \tag{20}$$

where I is the current intensity (A), E is the average voltage (V), t is the working time (h), and W is the adsorbent amount added to the slurry reactor (g).

### 3.5. Remediation Process Evaluation

The assessment of the regeneration capability was done by calculating the regeneration efficiency ($n_r$) (Equation (21)) [14].

$$n_r(\%) = \frac{Q_r}{Q_i} \cdot 100 \tag{21}$$

where $Q_r$ is the uptake after the EF regeneration and $Q_i$ is the initial uptake.

## 4. Conclusions

In this work, a catalytic adsorbent was simply synthesized by doping sepiolite with iron. This silicate-based material was selected due to its quick and efficient removal of [bpy] Cl, chosen as a model molecule of ILs whose treatment has been slightly reported. Further, [bpy] Cl adsorption on the obtained catalytic adsorbent (S-Fe) was conducted in a flow system, allowing the treatment of more than 4.5 L of inlet 200 mg L$^{-1}$ [bpy] Cl with only 30 g of S-Fe. Thus, S-Fe could be successfully used as an adsorbent at a real scale. This catalytic adsorbent was used as a heterogeneous catalyst in a subsequent EF regeneration process. It was recovered entirely under the optimal operational conditions. By using S-Fe as a catalytic adsorbent, two main problems can be avoided: the iron sludge generation in the EF reactor and the secondary pollution caused by the disposal of exhausted S-Fe. The catalytic adsorbent was used in multiple adsorption–EF regeneration cycles without damaging its

structure and functional groups during the process. This ensures the stability and the applicability of S-Fe as a potential adsorbent towards [bpy] Cl. However, the hereby studied IL has not been treated before and thus other studies with different adsorbents and regeneration processes may be evaluated in future studies.

**Supplementary Materials:** The following are available online at http://www.mdpi.com/2073-4344/10/9/950/s1. Figure S1. $N_2$ adsorption-desorption isotherms of sepiolite and S-Fe; Figure S2. Wide scan XPS spectra for (**a**) sepiolite and (**b**) S-Fe; Table S3. Comparison of the elemental composition between sepiolite and S-Fe; Figure S4. [bpy] Cl uptake profile in the five sections of the S-Fe adsorption column; Figure S5. Wide scan XPS spectra for (**A**) S-Fe, (**B**) exhausted S-Fe, (**C**) exhausted S-Fe after first EF treatment (**D**) exhausted S-Fe after second EF treatment, and (**E**) exhausted S-Fe after third EF treatment; Table S6. Elemental composition of the adsorbent during three adsorption-regeneration cycles.

**Author Contributions:** Conceptualization, M.Á.S., M.P. and A.M.D.; methodology, I.O. and A.M.D.; software, I.O.; validation, I.O.; formal analysis, M.P. and M.Á.S., investigation, M.Á.S., M.P. and A.M.D.; resources, M.Á.S. and M.P.; data curation, I.O. and A.M.D.; writing—original draft preparation, I.O. and A.M.D.; writing—review and editing, A.M.D., M.Á.S. and M.P.; visualization, M.Á.S. and M.P.; supervision, M.Á.S. and M.P.; project administration, M.Á.S. and M.P.; funding acquisition, M.Á.S. and M.P. All authors have read and agreed to the published version of the manuscript.

**Funding:** This work was funded by the Spanish Ministry of Science, Innovation and Universities and ERDF Funds (Projects CTM2017-87326-R and CTQ2017-90659-REDT). The authors would like to thank the Xunta de Galicia (project ED481B 2019/091) for financial support of A.M.D.

**Conflicts of Interest:** The authors declare no conflict of interest.

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
