# Peer review of "Iron-Loaded Catalytic Silicate Adsorbents: Synthesis, Characterization, Electroregeneration and Application for Continuous Removal of 1-Butylpyridinium Chloride"

_catalysts, doi:10.3390/catal10090950_

Round 1
Reviewer 1 Report
The paper deals with an interesting topic, catalytic adsorbent based on iron doped sepiolite for remediation of pollutants in water. In particular, the authors have investigated the efficiency of this catalyst in EF regeneration process. The paper is well structured and written. In my opinion it represent a significant contribution for the scientific community interested in water treatment. To sum up, I suggest the publication of this article in the present form.
Best regards.
Author Response
Thank you very much for your positive feedback

Reviewer 2 Report
From the humble point of view of the reviewer, this paper does not present enough quality to be published in Catalysts. My reasoning is the following:
1) Authors do not include previous works carry out in this research area. I can ensure that there are other similar works done, but authors do not include it because their results are worsen.
2) Materials and methods section is not well described. There are lacks regarding the equipment used and the exactly methodology followed.
3) The novelty of the work is questionable. Indeed, authors only reflect the novelty of their work in one text-line. A journal such as Catalysts deserves more novelty.
4) English language needs to be checked. Indeed there are several mistakes and some part of the text are hardly understandable.
Author Response
Reviewer’s comment 1) Authors do not include previous works carry out in this research area. I can ensure that there are other similar works done, but authors do not include it because their results are worsen.
Author’s response to the comment: We have now included actual references in section 2.7 which demonstrate this work defeats previous publications. We have not added a reference for the elimination of the same ionic liquid because the adsorption-regeneration hadn’t been reported before for this same IL.
Reviewer’s comment 2) Materials and methods section is not well described. There are lacks regarding the equipment used and the exactly methodology followed.
Author’s response to the comment Thanks to the reviewer’s comment the Materials and methods section has been modified so additional explanations have been added.
Reviewer’s comment 3) The novelty of the work is questionable. Indeed, authors only reflect the novelty of their work in one text-line. A journal such as Catalysts deserves more novelty.
Author’s response to the comment Considering the reviewer’s comment, we worried the novelty of the article was not properly justified. Thus, several sentences have been added which highlight again that 1) this specific IL has not been treated before under adsorption coupled to regeneration and 2) the usage of an iron polluted adsorbent for the subsequent Fenton-based regeneration treatment has not been so widely studied and hereby we propose a process that defeats some previous references.
Reviewer’s comment 4) English language needs to be checked. Indeed, there are several mistakes and some part of the text are hardly understandable.
Author’s response to the comment Thanks to the reviewer’s comment, English has been check by the authors and an English native colleague.

Reviewer 3 Report
This study compares Fe-sepiolite with sepiolite as adsorbents for a selected ionic liquid. While there is much information about these materials discussed and the motivation for conducting this study is sound, I felt that, as a reader, I wanted more information within the document itself. Specific comments are below:
1) Table 1 and the text on page 2 do not reflect each other.... The adsorption capacities in the text are on the order of 30 mg/g, whereas Table 1 lists them on the order of 0.4 mg/g. I am wondering if the table is listing adsorption capacities in different units ?
2) For the desorption studies, was the acetonitrile/water solution responsible for the 60+% desorption or was it the ammonium chloride responsible for the desorption? It seems much information could have been obtained decoupling these effects.
3) In Table 3, the kinetic model fit for graph (1c) is low (R^2 ~0.6)... however, on the graph, the fit appears very good...much more inline with the others (R^2 =~0.9+). Did you fit only a few points in the graph and then drew connecting lines in graph 1c?
4) In Table 5, the amount of Na2SO4 seems like it has the most impact on TOC removal in the regeneration solution. However, the table does not try Na2SO4 solution by itself to desorb the IL from the adsorbent. I think that this control (Na2SO4 with and without current) would have been enlightening. In addition, on the same table, the pH values are higher (8.5 vs 3) when varying the current. Therefore, there are really 2 variables here... pH and amperage to the regeneration system
5) In the conclusions, you indicate that because the technology was used in a flow system, the process could be used at high scale. In my opinion, this statement is subjective and not supported by the data presented. It is also not clear to me how the desorption/catalytic oxidation step using EF be incorporated into the column (in-situ) so that the adsorbent does not need to be removed from the column. Is there an effect of distance between the anode and cathode on the catalysis? Is the EF process scalable to large scale?
Author Response
Reviewer’s comment 1) Table 1 and the text on page 2 do not reflect each other.... The adsorption capacities in the text are on the order of 30 mg/g, whereas Table 1 lists them on the order of 0.4 mg/g. I am wondering if the table is listing adsorption capacities in different units ?
Author’s response to the comment: Thank you for your comment, the issue is 30 mg/g were attained on the isotherm, when using high ionic liquid concentrations, however, on the preliminary tests (Table 1) the initial effluent concentration was 10 mg/L (a low quantity to avoid unnecessary cost and to notice small differences). This fact has been now highlighted in the manuscript.
Reviewer’s comment 2) For the desorption studies, was the acetonitrile/water solution responsible for the 60+% desorption or was it the ammonium chloride responsible for the desorption? It seems much information could have been obtained decoupling these effects.
Author’s response to the comment: Thanks to the reviewer’s comment, we have performed the desorption experiences (Table 1) which lead us to the conclusion that the combination effect was the motor of the desorption capacity
Reviewer’s comment 3) In Table 3, the kinetic model fit for graph (1c) is low (R^2 ~0.6)... however, on the graph, the fit appears very good...much more inline with the others (R^2 =~0.9+). Did you fit only a few points in the graph and then drew connecting lines in graph 1c?
Author’s response to the comment: Thanks to the reviewer for the comment. On the manuscript we have explained better the fact that the intraparticule model fits badly (R2~0.6) but if different linear regions are defined, which are related to different limiting steps, R2 is around 0.9. We have added both graphs on supplementary material (Fig. S4).
Reviewer’s comment 4) In Table 5, the amount of Na2SO4 seems like it has the most impact on TOC removal in the regeneration solution. However, the table does not try Na2SO4 solution by itself to desorb the IL from the adsorbent. I think that this control (Na2SO4 with and without current) would have been enlightening. In addition, on the same table, the pH values are higher (8.5 vs 3) when varying the current. Therefore, there are really 2 variables here... pH and amperage to the regeneration system
Author’s response to the comment: Following the reviewer’s suggestion, several controls have been done. Thus, IL was monitored on the aqueous matrix as the degradation process happened, and no IL was detected on the effluent. This has been now added on the manuscript (section 2.6.1). Moreover, on the desorption tests (section 2.2), we have added the results after the usage of Na2SO4 as desorbing agent, which caused low IL desorption (8.3 %). Regarding the pH effect, AO was done under acid pH (3) and the degradation percentage was practically the same, demonstrating the fact that EF process has a better performance is due to the Fenton process, not because of the pH. This experience has been added to table 5.
Reviewer’s comment 5) In the conclusions, you indicate that because the technology was used in a flow system, the process could be used at high scale. In my opinion, this statement is subjective and not supported by the data presented. It is also not clear to me how the desorption/catalytic oxidation step using EF be incorporated into the column (in-situ) so that the adsorbent does not need to be removed from the column. Is there an effect of distance between the anode and cathode on the catalysis? Is the EF process scalable to large scale?
Author’s response to the comment: We are very grateful to the reviewer. Indeed, a new section (2.9. Future uses) has been added to answer these astute comments, which increases the soundness of the manuscript and opens a path for focused future researchers.

Reviewer 4 Report
In this manuscript, the authors described the synthesis, and application of iron loaded silicate adsorbents for removal of 1-butylpyridinium chloride.
This manuscript is rejected based on followings;
- The authors should provide information on why 1-butyl pyridinium chloride is required to be removed? How its presence affects the human health, ecosystem and environment?
- The other main reason is that as Sepiolite gives 100% removal then why iron loading was done? The authors must pay attention to the novelty and importance of the current research work.
- This manuscript should be corrected for the English language.
Author Response
Reviewer’s comment 1) The authors should provide information on why 1-butyl pyridinium chloride is required to be removed? How its presence affects the human health, ecosystem and environment?
Author’s response to the comment: We would like to highlight the introduction section to the reviewer where the toxic and persistent character of ionic liquids was pointed. Moreover, additional information has been added about the toxicity of ionic liquids and low biodegrability.
Reviewer’s comment 2) The other main reason is that as Sepiolite gives 100% removal then why iron loading was done? The authors must pay attention to the novelty and importance of the current research work.
Author’s response to the comment: The desorption from sepiolite was extremely difficult (60 %) even when adding a toxic organic solvent (acetonitrile) and a chloride salt (0.2 M). Thus, the regeneration of the adsorbent should be done by other means, as the adsorbent disposal is not an option we’d be happy with. The in situ regeneration using the EF process was a good option and as far as iron needs to be added to work as Fenton catalyst, it was added directly to sepiolite, to avoid all the problems associated to homogeneous catalyst. This is highlighted in the introduction and in sections 3.2.3 and 2.6.2.
Reviewer’s comment 3) This manuscript should be corrected for the English language.
Author’s response to the comment: Thanks to the reviewer comment, English has been checked by all authors and an English native speaker.

Reviewer 5 Report
This is an interesting study about the the application of iron loaded sepiolite as a catalytic adsorbent for the 1-butylpyridinium chloride treatment in an aqueous medium, focusing on the electrofenton, adsorption and fenton but some aspects should be revised.
Some numerical data could be introduced in the abstract, for example the kinetics for the best condition.
The introduction must contain more examples of low-cost materials for Fenton.
The application of Fenton for other processes should also be announced.
Table 1 should be revised the lines of different rows are not at the same position.
Why authors selected the sepiolite since NaY Zeolite and Attapulgite are very similar in terms of adsorption?
In the regeneration step what is the destination of the iron leached due to the regeneration?
The electroFenton presents best results than Fenton regarding to TOC removal, why?
An estimative of operation costs for this both processes could be addressed.
How many cycles it is possible to reuse this adsorbent?
Author Response
Reviewer’s comment 1) Some numerical data could be introduced in the abstract, for example the kinetics for the best condition.
Author’s response to the comment: Taking into consideration the reviewer’s comment, actual results have been included on the abstract
Reviewer’s comment 2) The introduction must contain more examples of low-cost materials for Fenton.
Author’s response to the comment: Following the reviewers’ suggestion, a few examples of low-cost adsorbents have been included on the introduction.
Reviewer’s comment 3) The application of Fenton for other processes should also be announced.
Author’s response to the comment: Following the reviewer’s comment, results reporting the usage of the Fenton process in aqueous solution for the degradation of a similar IL has been added on section 2.6.1.
Reviewer’s comment 4) Table 1 should be revised the lines of different rows are not at the same position.
Author’s response to the comment: Thanks to the reviewer’s comment, we have made columns wider to avoid the lines to be mixed depending on the word format
Reviewer’s comment 5) Why authors selected the sepiolite since NaY Zeolite and Attapulgite are very similar in terms of adsorption?
Author’s response to the comment: We are grateful for the reviewer’s comment. At the end of section 2.1 this fact was explained. We did the preliminary tests with a low IL concentration (10 ppm), thus indeed several adsorbents performed well. We increased then the IL concentration and whereas zeolite and attapulguite reached a maximum uptake of 7 mg/g, sepiolite attained 32 mg/g. This is why this latter was selected.
Reviewer’s comment 6) In the regeneration step what is the destination of the iron leached due to the regeneration?
Author’s response to the comment: Iron was measured throughout the experiences and it was not detected, so fortunately sepiolite is an extremely good adsorbent not only of IL but also of Iron and the organic degradation-focused processes seem to not affect its attachment. This fact has now been added in materials and methods and in results, in sections 2.6.1 and 2.6.2.
Reviewer’s comment 7) The electroFenton presents best results than Fenton regarding to TOC removal, why?
Author’s response to the comment: We thank the reviewer for his/her interest. This is caused because of the synergistic effect of the electroFenton process. A deeper explanation of the process synergy has now been added on section 2.6.2.
Reviewer’s comment 8) An estimative of operation costs for this both processes could be addressed.
Author’s response to the comment: Following the appropriate reviewer`s comment, a new section (2.9 future uses) has been added, and the economic evaluation and other considerations have been added.
Reviewer’s comment 9) How many cycles it is possible to reuse this adsorbent?
Author’s response to the comment: As it is depicted in section 2.8, we have performed 4 reuses without any detriment, so we strongly believe that the process can be used for many cycles.

Round 2
Reviewer 2 Report
Accept
Reviewer 4 Report
The authors revised and improved the manuscript. I accept this manuscript for publication.